# Dietary Rumen-Protected Taurine Enhances Growth Performance and Meat Quality in Heat-Stressed Crossbred Gan-Xi Goats via Modulating *GLUT4/PYGM*-Mediated Muscle Energy Metabolism

**DOI:** 10.3390/foods14193323

**Published:** 2025-09-25

**Authors:** Guwei Lu, Yijie Wang, Yuting Wei, Xin Liu, Siyu Lu, Xianghui Zhao, Qinghua Qiu, Mingren Qu, Lizhi Li, Yanjiao Li, Kehui Ouyang

**Affiliations:** 1Jiangxi Province Key Laboratory of Animal Nutrition/Animal Nutrition and Feed Safety Innovation Team, College of Animal Science and Technology, Jiangxi Agricultural University, Nanchang 330045, China; lgw0202020245@163.com (G.L.);; 2College of Life Science and Resources and Environment, Yichun University, Yichun 336000, China

**Keywords:** crossbred Gan-xi goats, serum cortisol, glycogen metabolism, textural properties, hepatic gluconeogenesis, muscle fatty acid profile

## Abstract

Heat stress induced by high temperature and humidity in southern China during summer reduce goat production efficiency and meat quality. Taurine (TAU), one of the most abundant amino acids in animal tissues, plays a vital role in alleviating heat stress and regulating energy metabolism through its involvement in glucose uptake and glycogen turnover. This study aimed to investigate the effects of rumen-protected (RP)-TAU on the meat quality, hepatic gluconeogenesis, and muscle energy metabolism of heat-stressed goats. During summer, twenty-four male crossbred Gan-xi goats (20.45 ± 2.95 kg) aged 5 months were randomly allocated to two groups treated with or without 0.4% RP-TAU (on a diet weight basis). After feeding for 60 days, six goats per treatment were slaughtered. Compared with the control group, RP-TAU supplementation significantly improved the growth performance of goats, as evidenced by increased final body weight, average daily gain, and average daily feed intake (*p* < 0.05). The goats in the RP-TAU group showed a reduced splenic index (*p* < 0.05), lower serum cortisol levels (0.05 < *p* < 0.1), and decreased muscle crude fat content (*p* < 0.01). Crucially, meat quality was improved with reduced hardness, gumminess, and chewiness (*p* < 0.05), indicating better textural properties. Nutritionally, RP-TAU supplementation modulated the muscle fatty acid profile, significantly reducing the concentrations of palmitic (a saturated fatty acid), palmitoleic (a monounsaturated fatty acid), and nervonic acids (*p* < 0.05), while cystine content was reduced (*p* < 0.05). RP-TAU supplementation significantly enhanced the muscle contents of glucose and glycogen, glycolytic potential, phosphofructokinase activity, and ATP level, while decreasing the pyruvate level and AMP/ATP ratio (*p* < 0.05). Gene expression analysis revealed the upregulation of *GLUT4* and *PYGM* and the downregulation of *GSK3β* in muscle (*p* < 0.05). These results indicated that dietary supplementation of RP-TAU might be beneficial to improve stress resistance and meat quality by increasing muscle energy supply and glucose uptake in Gan-xi goats.

## 1. Introduction

Gan-xi goats, which are particularly prevalent in the mountainous regions of western Jiangxi Province, China, boast notable characteristics such as high meat quality, strong adaptability, and excellent disease resistance. The meat from this breed is esteemed for its superior sensory attributes, including tenderness, a distinctive yet mild flavor, and juiciness, which have garnered significant consumer popularity. Annually, the production of these goats reaches 130,000, among which over 55,000 are breeding goats. Due to this considerable production scale, goats have played a key role in boosting local agricultural productivity significantly. Consequently, households in the area have seen their annual income rise by an average of more than CNY 10,000 [1]. However, western Jiangxi is a typical region with high temperatures and high humidity, where heat stress significantly impacts livestock production by impairing physiological and metabolic processes, leading to the reduced productivity and profitability of these goat farms. Excessive heat induces detrimental effects such as reduced feed intake, decreased metabolic rate, and altered energy metabolism [2,3]. Mechanistically, heat stress disrupts endocrine regulation by reducing circulating levels of thyroid hormones T3 and T4, resulting in declines in metabolic activity, dry matter intake (DMI), and production performance [4,5]. Additionally, heat stress significantly reduces growth performance and carcass characteristics, alters muscle energy metabolism, and downregulates genes linked to fat deposition and muscle development, thereby adversely affecting meat quality [6].

Dietary intervention serves as an effective strategy to modulate energy homeostasis, mitigate thermal stress, attenuate antioxidant depletion, and enhance meat quality in livestock subjected to heat stress [7]. Taurine (TAU, C_2_H_7_NO_3_S), with a molecular weight of 131.19, is a sulfur-containing amino acid that plays a crucial role in various physiological processes, such as energy metabolism and amino acid metabolism [8,9]. It also functions as a key precursor for bile acid metabolism and plays a significant role in lipid metabolism [10]. Studies have demonstrated that TAU supplementation enhances growth performance by increasing daily weight gain and improving feed efficiency in broilers [11]. TAU exhibits significant physiological benefits, primarily through its potent antioxidant properties. By mitigating oxidative stress and preserving cellular integrity across diverse conditions [12], TAU consequently enhances meat quality, notably improving color stability and water-holding capacity which is essential for freshness and palatability [12]. Beyond its protective roles, TAU plays a crucial part in energy metabolism. It facilitates glucose utilization, particularly under low-glucose conditions, by enhancing glycolytic enzyme activity [13,14]. Furthermore, TAU modulates lipid metabolism; under high-glucose or high-fat conditions, it downregulates lipogenic enzymes while enhancing fatty acid β-oxidation in hepatocytes [15]. Importantly, a key molecular mechanism underpinning TAU’s metabolic effects involves the activation of AMP-activated protein kinase (AMPK). This occurs via a calcium-dependent pathway involving CaMKK, leading to increased *GLUT4* expression and promoting glucose uptake in skeletal muscle cells. However, the information about the impacts of TAU on energy metabolism and whether TAU can improve growth performance and meat quality in heat-stressed ruminants, especially goats, is largely unknown. We hypothesize that TAU might affect growth performance and meat quality through influencing muscle energy metabolism and hepatic gluconeogenesis in goats suffering from heat stress. Therefore, it is necessary to carry out relevant research.

## 2. Materials and Methods

### 2.1. Animal Care

This study received ethical approval from the Animal Care and Use Committee of Jiangxi Agricultural University (Approval No. JXAULL-2024-07-05). All experimental procedures strictly adhered to the ARRIVE guidelines and relevant national standards for laboratory animal welfare.

### 2.2. Animal Treatments and Experimental Diets

Twenty-four male crossbred Gan-xi goats (5 months of age) with an average bodyweight of 20.45 ± 2.95 kg were randomly allocated to two groups with two RP-TAU supplemental levels (0 or 0.4% kg/diet, n = 12) in basal diets, and the experiment was conducted in Shangli County, Pingxiang City, Jiangxi Province, China, from 4 July 2024 to 2 September 2024. These groups were the HS group (heat stress + 0 kg RP-TAU in basal diet) and TAU group (heat stress + 0.4% kg/diet RP-TAU in basal diet), with twelve replicates in each treatment. After a 10-day pre-feeding period and a subsequent 60-day feeding phase, a total of 12 male crossbred Gan-xi goats (6 from each treatment), were selected for euthanasia and sample collection following a 12 h food withdrawal prior to slaughter. The goats were fed twice daily at 08:00 and 16:00, with water available ad libitum. All uneaten feed was removed and weighed before each morning feeding. The amount of feed offered was adjusted daily to ensure approximately 10% leftovers. Temperature data were collected using a temperature and humidity logger (Delixi Electric, Leqing, Zhejiang, China), positioned approximately 1.6 m above the ground, to record ET and RH at approximately 07:30, 14:30, and 22:30 daily. Figure 1 displays the THI during the experiment. The temperature–humidity index (THI) was computed using the established model described in Hamzaoui et al. [16]:THI = (1.8 × T_db_ + 32) − (0.55 − 0.0055 × RH) (1.8 × T_db_ − 26.8)
where T_db_ is the dry bulb temperature (°C) and RH is the relative humidity (%).

Taurine (TAU, purity ≥ 99.8%; Product Lot No. C002-M1809077) was purchased from Hubei Grand Life Science & Technology Co., Ltd. (Wuhan, Hubei, China). The rumen-protected TAU (RP-TAU) product was then produced by Hangzhou King Techina Feed Co., Ltd. (Hangzhou, China) using the aforementioned TAU as the core material. The TAU coating rate was 61.4%, the rumen bypass ratio (12 h) was 91.2%, and the dissolution of intestinal fluid (12 h) was 86.3%. The dosage of RP-TAU (0.4% of the diet) was selected based on previous studies reporting that this level of dietary taurine supplementation effectively improved heat-stress resistance and growth performance in livestock [17]. The formulation of the basal diet was designed to meet the nutritional requirements of growing meat goats, according to the Chinese Agricultural Industry Standard (Nutrient Requirements of Meat Sheep) (NY/T 816-2021) [18]. The composition and nutrient levels of the experimental diet are shown in Table 1.

### 2.3. Growth Performance

All goats were weighed following a 12 h fasting period at both the experimental start and conclusion. Cumulative feed consumption was monitored throughout the trial to calculate three key performance indices: average daily feed intake (ADFI), average daily weight gain (ADG), and feed-to-gain ratio (F/G). ADFI was computed as total feed intake over the trial duration divided by the number of experimental days. ADG was derived from the difference between final and initial body weights, divided by trial days. F/G was calculated as total feed intake divided by total body weight gain.

### 2.4. Sample Collection

Before slaughter, blood samples (15 mL) were collected from the jugular vein of goats using evacuated nonanticoagulative tubes. These blood samples were centrifuged at 3000× *g* for 10 min at 4 °C to isolate serum, which was then stored at −20 °C for subsequent analyses. Subsequently, the animals underwent slaughter via captive bolt stunning, followed by exsanguination, in accordance with animal welfare guidelines [19]. Immediately following slaughter, liver tissue and muscle samples (*M. longissimus thoracis* from the last rib of the left carcass) were collected and immersed in liquid nitrogen to preserve them for further evaluation of hepatic gluconeogenesis and muscle energy metabolism. At 30 min postmortem, additional *M. longissimus thoracis* (LT) samples were excised from the left side of the carcass. These samples were trimmed of external fat and connective tissue.

### 2.5. Meat Quality Measurements

Muscle samples were taken and lyophilized (BenchTop Pro desktop freeze dryer; Seattle, WA, USA) to determine the moisture content, and then crushed and sieved (40 mesh) for determination of the protein, fat, and ash contents [20,21,22]. The pH values of the LT muscle at 45 min post-slaughter (pH_45min_) and 24 h post-slaughter (pH_24h_) were measured using a pH meter equipped with an HI99163N electrode (Hanna Instruments, Padova, Italy). For each sample, measurements were taken three times across distinct regions, and the average value was recorded. To evaluate meat color, a 3nh colorimeter (Shenzhen, China) was used 45 min post-slaughter. After allowing freshly cut muscle surfaces to aerate for 20 min, the mean CIE L* (lightness), a* (redness), and b* (yellowness) values were determined at three different locations on each chop. For drip loss assessment, rectangular muscle strips (approximately 30 g) were excised along the muscle fiber orientation from the LT muscle after the carcasses had been stored at 4 °C for 24 h post-slaughter. These samples were weighed and suspended in polyethylene bags without contact with the bag material. After a 24 h period at 4 °C, the samples were reweighed to calculate drip loss. Texture profile analysis (TPA) was conducted on cooked LT muscle samples using a CT3 texture analyzer (Brookfield Engineering Laboratories Inc., Middleboro, MA, USA). At 24 h post-slaughter, muscle sections of approximately 2.5 cm in thickness were vacuum-sealed in individual polyethylene bags and cooked in a water bath at 80 °C until an internal temperature of 75 °C was reached. The cooked samples were then cooled to room temperature in running water, dried with absorbent paper to remove surface moisture, and cut into 1.0 cm thick slices to ensure a uniform surface for analysis. Three replicate measurements were performed per sample to quantify five textural parameters: hardness, cohesiveness, springiness, gumminess, and chewiness. The texture analyzer probe (TA3/100 configuration) was specifically programmed as follows: pre-test speed = 2 mm/s, test speed = 0.5 mm/s, post-test speed = 0.5 mm/s, target displacement = 10 mm, trigger threshold = 5 g [23].

### 2.6. Chemical Analyses

#### 2.6.1. Serum, Liver, and LT Muscle Pyruvate, Glucose, and Glycogen Determination

Pyruvate levels in serum and hepatic tissues were quantified spectrophotometrically using commercial kits (Jiancheng Bioengineering, Nanjing, China), following standardized protocols. Serum and hepatic glucose concentrations were assayed with a glucose oxidase-based kit from the same manufacturer. Hepatic and longissimus thoracis (LT) muscle glycogen content was determined via enzymatic hydrolysis kits (Jiancheng Bioengineering), adhering to the supplier’s specifications.

#### 2.6.2. Muscle Glycolytic Potential

A frozen LT muscle sample (0.5 g) was homogenized in 4.5 mL of ice-cold perchloric acid solution (0.85 M HClO_4_) using a high-speed homogenizer (Wuhan Servicebio Technology Co., Ltd., Wuhan, China) at 13,500 rpm for 30 s while maintained on an ice bath. The resulting suspension was centrifuged at 2700× *g* for 10 min at 4 °C to separate the supernatant and pellet fractions. The supernatant was neutralized by adding 10 M KOH, followed by enzymatic hydrolysis of glycogen into glucose using amyloglucosidase. Subsequently, glucose-6-phosphate in the supernatant was converted to glucose via glucose-6-phosphatase activity. Glucose concentrations were quantified with a glucose oxidase-based assay kit (Jiancheng Bioengineering Institute, Nanjing), following the manufacturer’s analytical protocols. The lactate content in the muscle tissue was analyzed using a diagnostic kit from Nanjing Jiancheng Bioengineering Institute, following the manufacturer’s instructions. Glycolytic potential (GP) was calculated as described by Monin and Sellier [24], with the following formula: GP = 2 × glycogen + lactate.

#### 2.6.3. Determination of Enzyme Activities

The activities of creatine kinase (CK) and lactate dehydrogenase (LDH) in blood serum were determined using commercial CK and LDH kits (Nanjing Jiancheng Bioengineering Institute), following the manufacturer’s instructions. For enzyme activity assays in liver or LT muscle, 0.5 g of frozen liver or LT muscle samples was homogenized in ice-cold physiological saline using a high-speed homogenizer for 1 min, followed by centrifugation at 2700× *g* for 10 min at 4 °C. Hepatic soluble fractions were assayed for phosphoenolpyruvate carboxykinase (PEPCK), glucose-6-phosphatase (G-6-pase), and pyruvate carboxylase (PC) activities. Parallel analysis of longissimus thoracis (LT) muscle cytosolic extracts quantified phosphofructokinase (PFK), hexokinase (HK), pyruvate kinase (PK), and lactate dehydrogenase (LDH) activities. Total protein concentration in LT muscle was colorimetrically determined with a BCA kit (TransGen Biotech, Beijing, China) following the standardized protocols of the manufacturer. The activities of PEPCK, G-6-pase, PFK, HK, PK, and LDH were measured using commercial kits; PEPCK, PFK, G-6-pase, HK, PK, and LDH kits were obtained from Nanjing Jiancheng Bioengineering Institute. PEPCK activity was excluded from normalization; for all other enzymes in liver or LT muscle samples, activities were standardized against the total protein concentration in LT muscle, following the manufacturer’s guidelines.

#### 2.6.4. Creatine and Phosphocreatine Determination

The contents of creatine and phosphocreatine in LT muscle were determined using high-performance liquid chromatography (HPLC) based on the methodology described by Li et al. [25], with modifications as follows. A total of 0.3 g of frozen LT muscle was ground into a fine powder and homogenized in 2 mL of ice-cold 5% perchloric acid using a high-speed homogenizer at 13,500 rpm for 30 s under constant cooling. The homogenate was then allowed to stand at 4 °C for 15 min prior to centrifugation at 10,000× *g* for 10 min. Following this, 900 μL of the supernatant was neutralized with an equal volume (900 μL) of 0.8 M K_2_CO_3_ and centrifuged again at 10,000× *g* for 15 min. The resulting supernatant was filtered through a 0.45 μm membrane filter before being injected into a Shimadzu LC-2030 Plus HPLC system (Shimadzu Corporation, Kyoto, Japan). The chromatographic separation was performed using an AQ-C18 column (250 mm × 4.6 mm I.D., 5 μm; Yuexu Co., Ltd., Shanghai, China). The mobile phase consisted of HPLC-grade methyl cyanide (mobile phase A) and a phosphate buffer (mobile phase B) containing 1.15 mM tetra-butylammonium hydrogen sulfate and 29.4 mM potassium dihydrogen orthophosphate (pH 5.1). The eluent was prepared as a mixture of 2% mobile phase A and 98% mobile phase B, with the following conditions: column temperature at 25 °C, injection volume of 20 μL, UV detection at 210 nm, flow rate of 1.0 mL/min, and total runtime of 20 min. The system was set to auto-inject mode for sample analysis.

#### 2.6.5. Adenosine Phosphates Analyses

The concentrations of adenosine phosphates (ATP, ADP, and AMP) in the LT muscle were determined using HPLC, with modifications based on the method described by Li et al. [25]. A fresh frozen LT muscle sample (0.5 g) was grounded into a fine powder and mixed thoroughly with 2.5 mL of cold 7% perchloric acid in an ice bath using a high-speed homogenizer at 13,500 rpm for 30 s. The mixture was then spun down at 15,000× *g* for 10 min at 4 °C to collect the supernatant. An 850 μL aliquot of the supernatant underwent neutralization with 850 μL of 0.85 M KOH solution. The mixture was then centrifuged (15,000× *g*, 4 °C, 10 min) to precipitate KClO_4_ residues. The clarified supernatant was filtered through a 0.45 µm microporous membrane before injection into the Shimadzu LC-2030 Plus HPLC system (Shimadzu, Kyoto, Japan). The chromatographic separation was performed using an AQ-C18 column (250 mm × 4.6 mm I.D., 5 μm; Yuexu Co., Ltd., Shanghai, China). The mobile phase system comprised two phases: Phase A (HPLC-grade methanol solvent) and Phase B (phosphate buffer solution). Phase B was formulated with 2.5 mM tetrabutylammonium hydrogen sulfate, 0.04 M potassium dihydrogen phosphate, and 0.06 M dipotassium hydrogen phosphate (pH 7.0), then sterilized via filtration through a 0.45 µm cellulose acetate membrane. The mobile phase ratio was set at 13.5% A to 86.5% B. The column temperature was maintained at 30 °C, and the injection volume was 10 µL with UV detection at 254 nm. The total run time per sample was 15 min, and the flow rate was set to 1.0 mL/min. Sample analysis was conducted using an auto-injection sequence. Peaks were identified and quantified based on standard calibration curves.

#### 2.6.6. Fatty Acid Analyses

The fatty acid composition of the LT muscle tissue was analyzed using gas Thermo Fisher Trace1310 ISQ gas chromatography–mass spectrometry system (Thermo Fisher Scientific, Waltham, MA, USA) equipped with a TG-5MS column (30 m × 0.25 mm × 0.25 µm), with modifications based on the method described by Zhang et al. [26]. Briefly, 2.0 g of tissue was homogenized in 10 mL of methanol/chloroform (2:1, *v*/*v*) containing 0.1% butylated hydroxytoluene (BHT) as an antioxidant. The mixture was vortexed for 30 s and then incubated at room temperature for 60 min. Following centrifugation at 4000 rpm for 10 min, the supernatant was collected and washed with 5 mL of 0.9% sodium chloride solution to remove any residual proteins or salts. The organic layer was separated, dried under nitrogen gas, and subjected to derivatization with N, O-bis(trimethylsilyl)acetamide (BSTFA) at 60 °C for 30 min. The temperature program for the analysis was as follows: initial heating at 80 °C for 1 min, followed by a gradual increase to 200 °C at a rate of 10 °C per minute; the oven continued to rise to 250 °C at 5 °C per minute and finally reached 270 °C at a rate of 2 °C per minute, maintaining this temperature for an additional 3 min.

#### 2.6.7. Amino Acid Analyses

The concentrations of amino acids in the LT muscle were determined using HPLC (Agilent Technologies Inc., Santa Clara, CA, USA) based on the methodology described by Zhang et al. [26], with modifications as follows. The ground samples were subjected to hydrolysis using HCl at 110 °C for 22 h. After cooling, the mixture was dried under vacuum and purged with nitrogen gas. Subsequently, phenyl isothiocyanate (PITC) derivatization solution was added, and the reaction proceeded at room temperature for 30 min before diluted with 0.45 mL of mobile phase A, thoroughly mixed, and injected into the HPLC system for analysis.

### 2.7. RNA Extraction and Real-Time Fluorescent Quantitative PCR Analysis

Total RNA was isolated from longissimus thoracis (LT) muscle and hepatic tissues with a commercial RNA extraction kit (Quanshijin Biotechnology, Beijing, China), adhering strictly to the supplier’s protocols. RNA concentration and purity were subsequently assessed spectrophotometrically using an applied biosystems by Thermo Fisher Scientific Quant Studio^TM^ 5 Real-Time PCR Instrument (Life Technologies Holdings Pte Ltd., Singapore). Only RNA samples with acceptable quality were used for subsequent experiments. Gene-specific primers were designed based on published gene sequences obtained from the NCBI database, using Primer Premier 5.0 software, and synthesized by Sangon Biotech (Shanghai, China).

Complementary DNA (cDNA) was synthesized from total RNA using reverse transcription in a 20 µL reaction system. The mixture was gently blended and incubated at 50 °C for 5 min, followed by heat inactivation at 85 °C for 5 s. cDNA samples were stored at −20 °C for further analysis. Preliminary dilution trials were conducted to determine the optimal dilution factor for qPCR. Quantitative real-time PCR (qPCR) was performed in a 20 µL reaction system using PerfectStart™ Dye-based qPCR Master Mix (TransGen Biotech, Beijing, China), according to the manufacturer’s instructions. The expression of target genes was normalized to β-actin as an internal reference. For LT muscle samples, the relative mRNA expression of *LKB1*, *AMPKα2*, *PYGM*, *GSK3β*, and *GLUT4* was analyzed. In liver samples, the expression of *LKB1*, *AMPKα2*, *PCK1*, and *PCK2* was determined. All reactions were conducted in triplicate. The relative quantification of gene expression was calculated using the 2^−ΔΔ*Ct*^ method.

### 2.8. Statistical Analysis

All data were analyzed using SPSS Statistics software (version 25.0, IBM, Armonk, NY, USA). The normality of the data distribution was tested using the Shapiro–Wilk test, and the homogeneity of variances was verified using Levene’s test. All data met the assumptions of normality and homogeneity of variances. Data from the two treatment groups were compared using an independent samples Student’s *t*-test. Results are presented as the mean and standard error of the mean (SEM). Differences were considered statistically significant at *p* < 0.05.

## 3. Results

### 3.1. Temperature–Humidity Index

As presented in Figure 1, during the period of the experiment, the THI index in the goat pen remained above 72. Among these, the THI exceeded 88 for 35 days, while the THI ranged between 79 and 88 for 25 days.

### 3.2. Growth Performance and Viscera Index

As presented in Table 2, there was no significant difference in initial body weight between the HS and TAU groups (*p* > 0.05). Moreover, the Gan-xi goats in the RP-TAU-supplemented group had a higher value of final body weight (29.7 vs. 32.5 kg; *p* < 0.05), ADG (0.15 vs. 0.19 kg; *p* < 0.05), and ADFI (1.06 vs. 1.19 kg; *p* < 0.05) than those in the HS group.

Compared with the HS group, the splenic index (0.16 vs. 0.12; *p* < 0.05) was significantly decreased in goats in the TAU group. Furthermore, there was no significant difference in the cardiac index, renal index, or hepatic index between the HS and TAU groups.

### 3.3. Serum Parameters

As presented in Table 3, the goats in the RP-TAU supplemented group had a tendency towards lower activities of serum cortisol (49.1 vs. 45.6 ng/mL; 0.05 < *p* < 0.1) than those in the HS group. However, RP-TAU supplementation did not significantly affect the activities of CK or LDH or the glucose content in the serum.

### 3.4. M. longissimus thoracis Composition and Meat Quality

As shown in Table 4, compared with the HS group, the percentage of crude fat (11.64 vs. 8.42%; *p* < 0.01) was significantly decreased in goats in the TAU group. Moreover, there was no significant difference in the percentage of muscle water, crude protein, or crude ash between the HS and TAU groups (*p* > 0.05).

Compared with the HS group, the hardness (400 vs. 187 kg; *p* < 0.05), gumminess (275 vs. 141 kg; 0.05 < *p* < 0.1), and chewiness (8.02 vs. 4.06 mJ; 0.05 < *p* < 0.1) of LT muscle were significantly decreased in the TAU group. However, there was no significant effect on the other meat quality characteristics.

### 3.5. Muscle Glycolytic Potential and Glycolytic Enzymes Activity

As shown in Table 5, compared with the HS group, the muscle glycogen content (2.93 vs. 5.32 μmol/g; *p* < 0.05) and GP (36.8 vs. 44.6 μmol/g; *p* < 0.01) were significantly increased in goats in the TAU group, while dietary RP-TAU supplementation increased the muscle glucose content (1.58 vs. 1.28 μmol/g; *p* < 0.05) and concentration of PFK (60.3 vs. 38.6 μmol/g) (*p* < 0.01) and decreased the concentration of muscle pyruvate (220 vs. 352 μmol/g prot; *p* < 0.05) of goats compared to those in the HS group. Moreover, there was no significant difference in the contents of muscle lactate or the enzyme activity of HK, PK, or LDH between the HS and TAU groups (*p* > 0.05).

### 3.6. Regulation of Muscular Phosphagen and Adenylate Systems

The results of muscle PCr, Cr, and adenosine phosphates are presented in Table 6. Compared with the HS group, TAU significantly decreased the muscle AMP/ATP (9.87 vs. 2.24; *p* < 0.05) in goats. Supplementation with RP-TAU increased the level of muscle ATP (97.7 vs. 131.6 µg/g; *p* < 0.01) and significantly decreased the level of muscle AMP (833 vs. 284 µg/g; 0.05 < *p* < 0.1). Furthermore, there was no significant effect on the other characteristics between the two groups.

### 3.7. Hepatic Gluconeogenesis

As shown in Table 7, compared with the HS group, dietary RP-TAU supplementation has no significant effect on any of hepatic gluconeogenesis parameters (*p* > 0.05).

### 3.8. M. longissimus thoracis Fatty Acids and Amino Acids

The results of muscle fatty acids are presented in Table 8. Compared with the HS group, the palmitic acid (171 vs. 107 mg/100 g; *p* < 0.05), palmitoleic acid (8.23 vs. 4.30 mg/100 g; *p* < 0.05), and nervonic acid (9.45 vs. 7.15 mg/100 g; *p* < 0.05) of LT muscle were significantly decreased in goats in the TAU group. However, there was no significant effect on the other fatty acid characteristics.

As detailed in Table 9, dietary taurine supplementation significantly reduced cystine content in caprine longissimus thoracis muscle (0.40 vs. 0.44 g/100 g in HS controls; *p* < 0.05), representing an 8.3% decrease. Conversely, the overall amino acid profile remained unaltered between the treatment groups (*p* > 0.05).

### 3.9. Real-Time PCR

The primer sequences of genes were shown in Table 10.

The expression levels of the *LKB1*, *AMPKα2*, *PYGM*, *GLUT4*, and *GSK3β* genes in the muscle of goats are presented in Figure 2. Compared with the HS group, the group supplemented with RP-TAU showed a significant increase in the expression of *GLUT4* and *PYGM*, and a decrease in the expression of *GSK3β* was observed in goat muscle (*p* < 0.05). No significant differences were found in the mRNA expression levels of *LKB1* and *AMPKα2* in the muscle of goats among the different treatment groups (*p* > 0.05).

The expression levels of the *LKB1*, *AMPKα2*, *PCK1*, and *PCK2* genes in the liver of goats are presented in Figure 3. Compared with the HS group, the expression of *LKB1* was significantly increased in goats in the TAU group. Moreover, there was no significant difference in the liver expression of the *AMPKα2*, *PCK1*, and *PCK2* genes between the HS and TAU groups (*p* > 0.05).

## 4. Discussion

This study aimed to investigate the effects of heat stress on growth performance, blood indices, meat quality, and energy metabolism in crossbred Gan-xi goats. We focused on exploring whether RP-TAU could alleviate the adverse impacts of heat stress on energy metabolism and meat quality. The systemic energy crisis induced by heat stress is central to these impacts. To increase heat dissipation, animals reduce metabolic efficiency. This leads to a drop in ATP availability, which propagates through reduced feed intake, glycogen depletion, oxidative damage, and ultimately poorer growth and meat quality [27,28,29].

It is well-documented that heat stress reduces daily feed intake to lower metabolic heat production. Concurrently, it aggravates gastrointestinal dysfunction, oxidative stress, endocrine disruption, and metabolic remodeling. These changes compromise energy and nutrient absorption, ultimately reducing weight gain efficiency [27,30,31]. However, TAU ameliorated growth performance in heat-stressed broilers by reducing serum leptin levels, increasing triiodothyronine (T_3_) concentrations, and downregulating hypothalamic *POMC* and *LEPR* expression, thereby alleviating appetite suppression [32]. From the results of this study, it is evident that RP-TAU significantly improved feed intake and average daily gain in goats under heat-stress conditions. This finding is consistent with the work by Li et al. [15], who reported similar effects when combining folic acid and TAU in heat-stressed lambs. However, Chen et al. [33] observed no significant changes in production performance when adding 20 g/day RP-TAU to the diet of high-altitude Yaks, which may be attributed to differences in dosage.

Beyond growth performance, we also observed changes in visceral indices. Studies have demonstrated that alterations in visceral indices can serve as indicators of an animal’s adaptation to environmental stressors or dietary interventions [34]. In this study, goats supplemented with RP-TAU exhibited a significant reduction in spleen index, while no notable changes were observed in other visceral indices. We interpret this splenomegaly within an energy-crisis framework, viewing it as collateral inflammatory damage driven by ROS overproduction. RP-TAU’s ability to curb this damage corroborates its upstream role in restoring mitochondrial efficiency and lowering oxidative load. This reduction in spleen pathology is a critical early indicator of RP-TAU’s systemic anti-stress and antioxidant efficacy, which likely underpins subsequent improvements in metabolic function. This view is supported by Chen et al. [35], who revealed that chronic heat stress induces splenomegaly in broilers. Their study showed that heat stress triggers oxidative stress, upregulates pro-inflammatory cytokines (IL-6, TNF-α, IFN-γ), downregulates the anti-inflammatory cytokine IL-4, and activates the TLRs/MyD88/NF-κB pathway. This suggests that under heat-stress conditions in our experiment, heat stress may have caused splenic damage or edema leading to physiological enlargement of the spleen, while dietary supplementation with RP-TAU significantly reduced the spleen index. This reduction in spleen index strongly suggests that RP-TAU alleviated heat-stress-induced splenic pathology, potentially through its recognized antioxidant and immunomodulatory properties. TAU enhances spleen health by improving immune function, as evidenced by elevated Th1/Th2 cytokine levels and reduced chemotherapy-induced immunosuppression [36]. Unfortunately, we did not collect samples from the spleen in this study, and further research is needed to elucidate the specific mechanisms underlying the reduction in spleen index caused by RP-TAU supplementation.

Crucially, this observed mitigation of stress-associated organ changes (spleen index) aligns with and is likely functionally linked to the systemic stress reduction indicated by serum cortisol levels. Heat stress induces a series of stress responses in the organism, among which cortisol, a critical stress hormone, is significantly elevated in secretion under such conditions [37]. In the present study, reduced serum cortisol activity was observed in goats supplemented with RP-TAU, directly confirming RP-TAU’s efficacy in ameliorating the core physiological stress response under heat stress. This alleviation of systemic stress is a fundamental step in counteracting the negative impacts of heat stress. It suggests that RP-TAU may alleviate heat stress and mitigate its associated physiological responses in these animals. Importantly, this reduced systemic stress burden, evidenced by lower cortisol and a normalized spleen index, creates a more favorable internal environment. This environment is a critical prerequisite for restoring cellular energy homeostasis, which underpins the metabolic improvements we observed in muscle tissue. However, it should be noted that RP-TAU supplementation had no significant effect on serum CK, LDH activities, and glucose content in this study. This could be attributed to either the minimal impact of heat stress on these parameters under our experimental conditions or the intricate mechanisms through which RP-TAU affects these biochemical markers, which require further investigation.

Concurrently, RP-TAU exerted significant effects on muscle composition and texture, contributing directly to improved meat quality of heat-stressed goats. Specifically, RP-TAU supplementation significantly decreased the crude fat and texture characteristics (hardness, gumminess, chewiness) in muscle. The reduction in muscle hardness, gumminess, and chewiness suggests improved tenderness, which may be mechanistically linked to the observed decrease in shear force.

The reduction in intramuscular fat is a key mechanism behind these improvements. Chronic heat stress leads to a reduction in subcutaneous fat and a decline in intramuscular fat (IMF) content, which may enhance the animal’s heat dissipation efficiency [38]. Additionally, well-documented studies have demonstrated that TAU exhibits significant effects in reducing fat deposition and adipose tissue accumulation. TAU ameliorates lipid deposition by enhancing hormone-sensitive lipase (HSL)-mediated lipolysis to promote adipose tissue mobilization. Simultaneously, it suppresses acetyl-CoA carboxylase (ACC) and fatty acid synthase (FAS) expression to inhibit de novo lipogenesis [39]. This metabolic reprogramming is further amplified by the activation of mitochondrial carnitine palmitoyl l transferase 1 (M-CPT1), which drives β-oxidation efficiency, thereby establishing a catabolic-dominant lipid homeostasis that reduces ectopic lipid accumulation under metabolic stress conditions [40,41]. Therefore, the observed decrease in intramuscular crude fat content, driven by RP-TAU’s lipid-lowering effects, is one of the primary factors contributing to the improved tenderness (reduced hardness, gumminess, chewiness) of the meat. To date, few studies have reported the role of TAU in reducing shear force in ruminant meat. However, in swine nutrition, dietary TAU supplementation improves pork quality by enhancing oxidative fiber-related gene expression, improving mitochondrial biogenesis and function (via increased mtDNA content and ATP synthesis), and promoting myofiber remodeling from glycolytic to oxidative fibers through the calcineurin/nuclear factor of activated T cells c1 (CaN/NFATc1) signaling pathway [12]. Therefore, the observed decrease in intramuscular crude fat content, driven by RP-TAU’s lipid-lowering effects, is a primary factor contributing to the improved tenderness (reduced hardness, gumminess, and chewiness) of the meat.

A pivotal finding of this study, central to mitigating the downstream meat quality defects, is that RP-TAU profoundly improved muscle energy metabolism. It is well known that heat stress during pre-slaughter can lead to the depletion of glycogen stores in the muscle, thereby limiting ante-/postmortem glycolytic metabolism associated with the production of reactive oxygen species (ROS), and resulting in a high pHu and lower consumer acceptability [42,43,44]. A key finding of this study is that RP-TAU profoundly improved muscle energy metabolism, which is critical for combating heat-stress-induced meat quality defects. In this study, RP-TAU supplementation significantly increased the content of glycogen, glucose, GP, and PFK and decreased the content of pyruvate and ratio of AMP/ATP. This collective pattern indicates enhanced substrate availability (glycogen, glucose), accelerated glycolytic flux (increased PFK activity and GP), and improved cellular energy status (higher ATP, lower AMP/ATP ratio, reduced pyruvate accumulation). This optimized energy metabolism is the biochemical cornerstone for preventing high pHu and ensuring favorable postmortem meat development.

The enhancement of glycolytic function is likely mediated by the antioxidant properties of TAU. Heat stress suppresses PFK activity by inducing oxidative stress, which causes structural and functional damage to PFK via attacks from reactive oxygen species on its active center [45,46,47]. Our results indicate that RP-TAU supplementation effectively enhances the enzyme activity of PFK and glycolytic potential, which might be attributed to the antioxidant properties of TAU. TAU protects PFK from oxidative damage by scavenging free radicals with its antioxidant properties and also inhibits the HIF-1α signaling pathway to reduce PFK expression, thereby suppressing glycolysis [48]. The decline in pyruvate concentrations may be attributable to TAU’s facilitation of glycolytic intermediate utilization, potentially by mitigating intracellular calcium ion deficiency, stimulating GSH biosynthesis, and enhancing TCA cycle anaplerosis [49].

RP-TAU also positively influenced the phosphagen system. We also examined the contribution of the phosphagen system. In early postmortem metabolism, ATP levels are maintained by both this system and glycolysis [50]. In the present investigation, RP-TAU supplementation was found to augment the concentration of creatine within the phosphagen system (from 4.30 to 6.80 µmol/g, *p* = 0.088), while simultaneously increasing ATP levels and reducing AMP accumulation, thereby resulting in a decreased AMP/ATP ratio. Although existing research has not yet definitively established a direct correlation between TAU supplementation and enhanced phosphagen system function, prior findings have implicated several potential mechanisms. These include the antioxidant properties of TAU, its regulatory influence on calcium ion concentrations, and its role in preserving intracellular acid–base homeostasis [49,51,52]. In summary, these mechanisms are theorized to collectively augment energy metabolic efficiency. This enhancement ensures sufficient intracellular ATP supply, promotes prompt AMP clearance, minimizes pyruvate buildup, and optimizes muscle tissue energy status, particularly under heat-stress conditions. Critically, this restoration of muscle energy metabolism, particularly the preservation/restoration of glycogen stores and high ATP levels, provides the essential biochemical foundation for mitigating the risk of a high pHu value and is intrinsically linked to the observed improvements in meat texture and quality.

To understand the source of improved muscle glucose/glycogen, we examined hepatic gluconeogenesis, the primary glucose source in ruminants. In ruminants, hepatic gluconeogenesis is responsible for 80% of endogenous glucose synthesis, primarily utilizing propionate, glucogenic amino acids, lactate, glycerol, and pyruvate as substrates [53]. Previous research has demonstrated that TAU supplementation exerts no significant influence on hepatic gluconeogenesis during endurance exercise, as evidenced by non-significant changes in hepatic glycogen content, glucose-6-phosphatase (G6Pase) activity, and gluconeogenic amino acid concentrations [54]. Similarly, our results showed that RP-TAU supplementation has no significant effect on hepatic gluconeogenesis. This lack of effect on hepatic glucose production further underscores that the improvements in muscle energy metabolism and systemic stress response are the primary pathways through which RP-TAU exerted its beneficial effects in heat-stressed goats, rather than via enhanced hepatic glucose output.

We further investigated the LKB1/AMPK pathway in the liver. Consistent with these findings, our real-time PCR results demonstrated a significant increase in LKB1 expression with RP-TAU supplementation. Previous studies have demonstrated that pronounced physiological perturbations induced by stress can lead to ATP depletion and an elevated AMP/ATP ratio, subsequently activating AMP-activated protein kinase (AMPK) [55]. Activated AMPK modulates the promoter activity of the glucose-6-phosphatase (G6Pase) gene and the transcriptional expression of phosphoenolpyruvate carboxykinase (PEPCK) in hepatic cells, suggesting that AMPK activation may regulate gluconeogenic pathways through the transcriptional control of key enzyme-encoding genes [56]. Moreover, in this study, TAU supplementation markedly upregulated hepatic *LKB1* gene expression in heat-stressed goats. However, as a canonical upstream kinase, *LKB1* failed to activate its downstream target (*AMPK*), thereby failing to induce the transcription of phosphoenolpyruvate carboxykinase 1 (*PCK1*) and *PCK2*. This observation suggests that alternative LKB1-mediated signaling routes may be implicated in the metabolic modulation exerted by TAU. TAU serves as a critical substrate in bile acid metabolism and regulates this process by activating the *LKB1–AMPK* signaling axis, which facilitates the polarized trafficking of the bile acid transporter ABCB11 to the canalicular membrane [57]. Additionally, TAU promotes hepatocyte polarization via the cAMP–Epac–MEK–LKB1–AMPK cascade, thereby contributing to the morphogenesis and maintenance of the bile canalicular network and supporting normal hepatic physiological functions [58]. Overall, TAU supplementation did not induce hepatic expression of *PCK1* and *PCK2*, indicating no enhancement of gluconeogenesis based on hepatic gluconeogenic markers or gene expression.

Examining potential signaling pathways involved in the metabolic improvements, we focused on the LKB1/AMPK axis—a master regulator of cellular energy sensing. The LKB1/AMPK signaling cascade is a critical regulator of cellular energy homeostasis in animals. *LKB1*, an upstream kinase of *AMPK*, activates the *AMPKα2* subunit via phosphorylation at threonine 172 [57]. Alterations in AMPK activity are known to influence glycogen metabolism; *AMPK* activation inhibits glycogen synthase and promotes glycogen phosphorylase activity, collectively reducing intramuscular glycogen content [59,60]. In the present study, RP-TAU supplementation did not induce significant changes in *LKB1/AMPK* signaling gene expression compared to the HS group.

Despite the lack of change in the core pathway, key downstream metabolic effectors were modulated. However, despite the lack of change in core LKB1/AMPK transcripts, RP-TAU induced significant downstream effects on key metabolic regulators: it downregulated *GSK3β* mRNA levels while significantly upregulating *GLUT4* and *PYGM* transcription. These specific gene expression changes (downregulated *GSK3β*, upregulated *GLUT4* and *PYGM*) align well with the observed metabolic improvements: reduced *GSK3β* (a negative regulator of glycogen synthase) could favor glycogen storage, while increased *GLUT4* enhances glucose uptake and upregulated *PYGM* (glycogen phosphorylase muscle isoform) promotes glycogen breakdown when energy is needed. By enhancing *GLUT4* expression and function, TAU supplementation effectively improves glucose uptake and insulin resistance, providing a mechanistic basis for the beneficial effects of RP-TAU observed in this study.

This suggests an alternative signaling mechanism for RP-TAU. This precise gene expression pattern provides a compelling mechanistic narrative: increased *GLUT4* enhances glucose uptake [61,62], upregulated *PYGM* promotes glycogen breakdown to meet energy demands, and downregulated *GSK3β* (a negative regulator of glycogen synthase) potentially favors glycogen storage. It is noteworthy that this metabolic reprogramming occurred independently of the canonical *LKB1/AMPK* pathway, for which we detected no significant changes in gene expression. This suggests that RP-TAU modulates the **GLUT4/PYGM/GSK3β** network through alternative signaling routes. This pattern supports the enhanced glycolytic potential and energy status observed in muscle. *GSK3β* serves as both a critical regulator of glycogen synthesis within the *AMPK* axis and a pivotal downstream effector of phosphoinositide 3-kinase (*PI3K*), orchestrating diverse intracellular signaling networks [63,64]. Chen et al. [65] reported that TAU inhibits colorectal cancer cell invasion and metastasis by modulating the *AKT/GSK3β* pathway in SW480 and HT29 colorectal cancer cells. The observed downregulation of *GSK3β* and upregulation of *PYGM* mRNA expression may be attributable to AMPK-mediated regulatory responses under chronic heat-stress conditions. *AMPK* potentially mitigates glycogen synthesis via *GSK3β* inhibition, promotes glycogenolysis through *PYGM* activation to boost ATP generation, and enhances glucose uptake via *GLUT4*, collectively supporting rapid cellular energy replenishment. Thus, while the canonical *LKB1/AMPK* pathway was not fully activated in liver, specific elements of related signaling networks in muscle (*GSK3β*, *GLUT4*, *PYGM*) appear modulated by RP-TAU, contributing to the improved energy metabolism phenotype.

The lipid-lowering effect of RP-TAU was further reflected in the composition of muscle fatty acids. The relationship between muscle fatty acid composition and fat deposition constitutes a sophisticated physiological interplay. In the current study, RP-TAU supplementation decreased the content of almost all fatty acids per gram of LT muscle in goats, particularly palmitic acid, palmitoleic acid, oleic acid, and nervonic acid. This global reduction in fatty acid content is a direct consequence of the significantly decreased intramuscular crude fat (as discussed in the meat quality section), driven by RP-TAU’s established lipid-lowering mechanisms. This effect may result from TAU’s lipid-lowering properties, which led to a reduction in fat content in the LT muscle following TAU supplementation. As discussed earlier, TAU significantly attenuates adipose tissue accumulation primarily via downregulation of FAS expression to attenuate lipogenesis, while upregulating *ATGL* expression to enhance lipolysis [39]. Furthermore, the metabolite of TAU (N-acetyl taurine) engages the GFRAL receptor to modulate energy homeostasis, thereby diminishing adiposity [41]. This further explains the observed finding that the fatty acid content in the LT muscle of equal mass was significantly lower in the TAU-supplemented group compared to the control group. Previous research demonstrated that TAU supplementation exerts no significant influence on fatty acid composition or oxylipin generation in humans or cultivated hepatocytes, as evidenced by unaltered plasma and cellular levels of eicosapentaenoic acid, docosahexaenoic acid, and their derived oxylipins following TAU intake [66]. Nevertheless, the underlying mechanisms responsible for the marked decrease in specific fatty acids warrant further elucidation.

Amino acids serve as the fundamental structural components of proteins, playing essential roles in protein biosynthesis and modulating diverse physiological processes [67]. Optimal amino acid nutrition is vital for sustaining caprine health and enhancing production performance. Elevated concentrations of essential amino acids (EAAs) in muscular tissue are positively correlated with enhanced meat nutritional quality, while umami amino acids (UAAs) critically determine organoleptic properties and mediate flavor development in diverse food matrices [68]. Consequently, differential thermal degradation patterns of amino acids during cooking yield characteristic flavor profiles [69]. TAU, as a functional amino acid, plays a crucial role in the metabolic regulation of various amino acids, including cysteine, thereby modulating both the levels and metabolic pathways of amino acids in muscle tissue [70,71].

In this study, supplementation with RP-TAU led to a reduction in cystine content, without significantly affecting the levels of UAAs, EAAs, or total amino acids (TAAs). While this specific change (cystine reduction) occurred, the overall amino acid profile essential for meat nutritional quality and flavor potential remained largely unaltered by RP-TAU supplementation in this goat model, suggesting that the primary drivers of improved meat quality were the changes in energy metabolism, fat content, and texture, rather than major shifts in the amino acid pool. The study of Zhou et al. [71] demonstrated that a 0.3% TAU supplementation significantly enhanced the TAAs content in juvenile fish muscle, particularly elevating both essential and non-essential amino acids (NEAAs). This finding suggests that TAU facilitates amino acid deposition in muscle. However, species-specific differences may explain the lack of amino acid accumulation in goat muscle and the observed decrease in cystine levels. TAU synthesis primarily occurs via the transsulfuration pathway, where cysteine is first catalyzed by cysteine dioxygenase (CDO) to form cysteine sulfinic acid, which is subsequently converted into TAU [72,73,74]. Research indicates that TAU synthesis is influenced not only by cysteine levels but also by methionine metabolism. Methionine is converted into cysteine via homocysteine, thereby indirectly influencing TAU synthesis [75,76]. Therefore, the reduction in cystine observed here may reflect altered flux through the TAU synthesis pathway in response to exogenous RP-TAU supplementation, potentially sparing cysteine or modulating methionine metabolism, without detrimentally impacting the broader amino acid profile relevant to meat quality.

## 5. Conclusions

In conclusion, dietary RP-TAU supplementation effectively enhanced growth performance (increased ADFI, ADG), stress resistance (reduced cortisol, normalized splenic index), and meat quality (reduced fat, enhanced tenderness/texture) in heat-stressed crossbred Gan-xi goats. These improvements are attributed to RP-TAU’s multifaceted actions: (1) optimization of muscle energy metabolism, involving enhanced glucose uptake (*GLUT4*) and increased ATP production, mitigating meat quality defects like high ultimate pH (pHu); (2) amelioration of systemic stress responses, indicated by reduced cortisol, normalized splenic architecture, and protection of glycolytic enzyme activity (e.g., PFK). Critically, RP-TAU primarily targeted these pathways, with no significant impact on hepatic gluconeogenesis observed.

## Figures and Tables

**Figure 1 foods-14-03323-f001:**
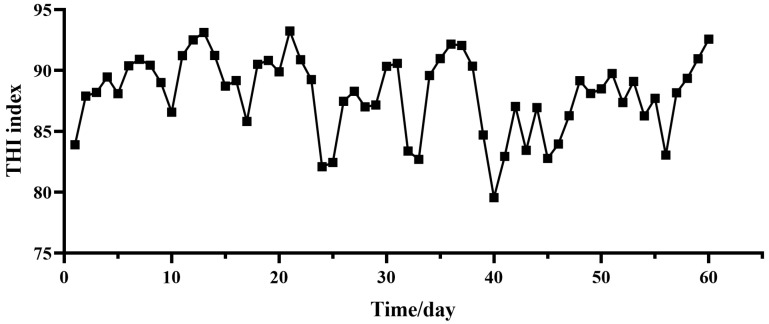
Temperature–humidity index (THI) during test period. Note: THI = (1.8 × T_db_ + 32) − (0.55 − 0.0055 × RH) (1.8 × T_db_ − 26.8). T_db_ = dry bulb temperature (°C); RH = relative humidity (%).

**Figure 2 foods-14-03323-f002:**
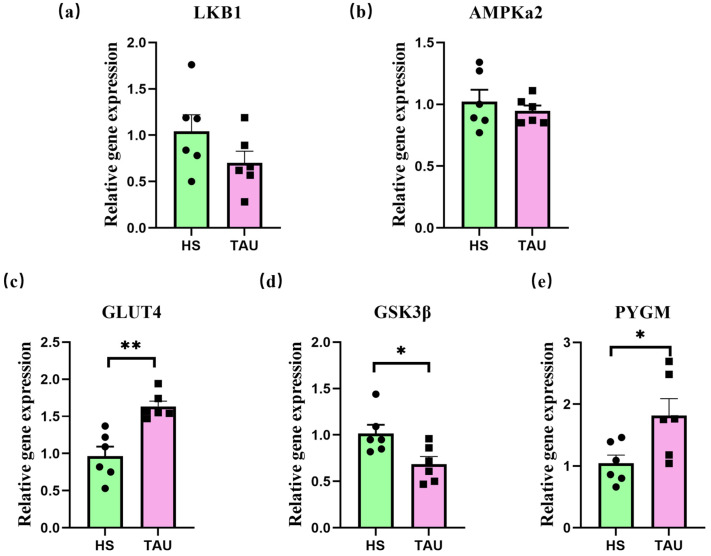
Effects of RP-TAU on *LKB1* (**a**), *AMPK* (**b**), *GLUT4* (**c**), *GSK3β* (**d**) and *PYGM* (**e**) gene expression in *M. longissimus thoracis* of heat-stressed crossbred Gan-xi goats. Note: HS = 0 kg RP-TAU in basal diet under heat stress; TAU = 0.4% kg/diet RP-TAU in basal diet under heat stress. Values shown represent the mean ± SE (n = 6). * *p* < 0.05, ** *p* < 0.01. Results were analyzed using Dunnett’s test.

**Figure 3 foods-14-03323-f003:**
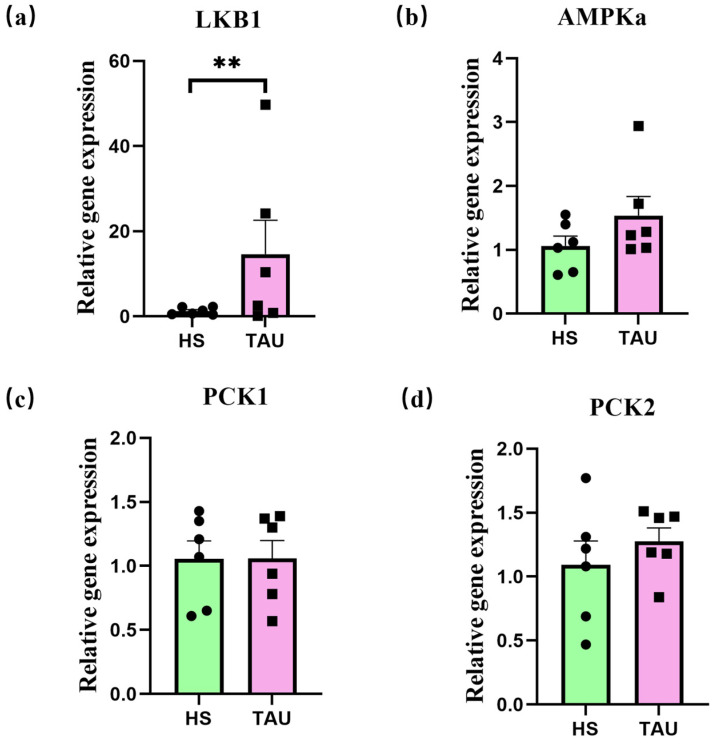
Effects of rumen-protected TAU on *LKB1* (**a**), *AMPK* (**b**), *PCK1* (**c**), and *PCK2* (**d**) gene expression in liver of heat-stressed crossbred Gan-xi goats. Note: HS = 0 kg RP-TAU in basal diet under heat stress; TAU = 0.4% kg/diet RP-TAU in basal diet under heat stress. Values shown represent the mean ± SE (n = 6). ** *p* < 0.01.

**Table 1 foods-14-03323-t001:** Composition and nutrient levels of experimental diet (air-dry basis, %).

Ingredients	Content (%)
Wheat straw	40.00
Corn	30.00
Wheat bran	9.84
Soybean meal	15.00
Sodium bicarbonate	0.78
Salt	1.38
Premix ^1^	3.00
Total	100
**Nutrient levels ^2^**	**Content**
Dry matter	92.03
ME/(MJ/kg)	10.88
Crude protein	16.02
Crude fat	4.92
Neutral detergent fiber	34.58
Acid detergent fiber	16.12
Crude ash	10.53

Note: ^1^ The premix (per kg of diet) is: 200,000 IU of vitamin A, 30,000 IU of vitamin D3, 600 mg of vitamin E, 600 mg of Mn, 1000 mg of Zn, 50 mg of Cu, 2 mg of Se, 10 mg of Co, 6 mg of I, 10% of Ca and 3% of P. ^2^ Nutrient levels were analyzed values, while ME was a calculated value.

**Table 2 foods-14-03323-t002:** Effects of rumen-protected taurine on the growth performance and viscera index of heat-stressed crossbred Gan-xi goats.

Items	Groups ^1^		SEM	*p*-Value
	HS	TAU		
Initial Body Weight (kg)	20.6	21.0	0.800	0.642
Final Body Weight (kg)	29.7	32.5	1.160	0.027
ADG (kg/d)	0.15	0.19	0.014	0.012
ADFI (kg/d)	1.06	1.19	0.047	0.011
F/G	6.78	6.09	0.358	0.069
Cardiac Index	0.38	0.39	0.019	0.820
Splenic Index	0.16	0.12	0.014	0.043
Hepatic Index	1.68	1.71	0.128	0.819
Renal Index	0.34	0.33	0.012	0.654

Note: ^1^ HS = 0 kg RP-TAU in basal diet under heat stress; TAU = 0.4% kg/diet RP-TAU in basal diet under heat stress. ADG = average daily gain; AFI = average daily feed intake; F/G = feed intake/weight gain. SEM = standard error of the mean.

**Table 3 foods-14-03323-t003:** Effects of rumen-protected taurine on the serum indicators of heat-stressed crossbred Gan-xi goats.

Items	Groups ^1^		SEM	*p*-Value
	HS	TAU		
Cortisol (ng/mL)	49.1	45.6	1.828	0.081
CK (U/L)	884	773	322.2	0.741
LDH (U/L)	435	421	35.19	0.708
Glucose (mmol/L)	5.37	4.59	1.029	0.465

Note: ^1^ HS = 0 kg RP-TAU in basal diet under heat stress; TAU = 0.4% kg/diet RP-TAU in basal diet under heat stress. SEM = standard error of the mean; CK = creatine kinase; LDH = lactate dehydrogenase.

**Table 4 foods-14-03323-t004:** Effects of RP-TAU on the *M. longissimus thoracis* composition and meat quality of heat-stressed crossbred Gan-xi goats.

Items	Groups ^1^		SEM	*p*-Value
	HS	TAU		
Moisture (%)	75.8	75.9	0.209	0.653
Crude protein (%)	86.2	87.7	3.188	0.644
Crude fat (%)	11.6	8.42	0.815	0.003
Crude ash (%)	4.00	3.96	0.163	0.835
pH_45min_	6.49	6.40	0.577	0.162
pH_24h_	5.72	5.69	0.044	0.543
Lightness (L*)	35.6	33.3	1.115	0.808
Redness (a*)	9.04	8.52	1.039	0.597
Yellowness (b*)	6.71	6.89	0.367	0.644
Drip loss (%)	2.10	2.55	0.660	0.522
Cooking loss (%)	10.9	14.2	1.981	0.123
Hardness (g)	400	187	77.91	0.049
Cohesiveness	0.40	0.45	0.027	0.111
Springiness (mm)	2.90	2.99	0.137	0.558
Gumminess (g)	275	141	27.31	0.063
Chewiness (mJ)	8.02	4.06	1.702	0.069

Note: ^1^ HS = 0 kg RP-TAU in basal diet under heat stress; TAU = 0.4% kg/diet RP-TAU in basal diet under heat stress. RP-TAU = rumen-protected taurine; SEM = standard error of the mean.

**Table 5 foods-14-03323-t005:** Effects of RP-TAU on glycolytic parameters and enzyme activities in *M. longissimus thoracis* of heat-stressed crossbred Gan-xi goats.

Items	Groups ^1^		SEM	*p*-Value
	HS	TAU		
Glycogen (μmol/g)	2.93	5.23	0.989	0.043
Glucose (μmol/g)	1.28	1.58	0.131	0.044
Pyruvate (μmol/gprot)	352	220	55.07	0.038
Lactate (μmol/g)	27.7	31.7	2.668	0.160
GP (μmol/g)	36.8	44.6	2.042	0.003
HK (U/mgprot)	0.44	0.70	0.143	0.104
PFK (U/mgprot)	38.6	60.3	6.889	0.010
PK (U/gprot)	1.45	0.98	2.718	0.140
LDH (U/kgprot)	1.38	1.33	0.092	0.637

Note: ^1^ HS = 0 kg RP-TAU in basal diet under heat stress; TAU = 0.4% kg/diet RP-TAU in basal diet under heat stress. Glycolytic parameters include glycogen content, lactate content, and glycolytic potential. SEM = standard error of the mean; GP (glycolytic potential) = 2 × (glycogen) + lactate [24]; RP-TAU = rumen-protected taurine; LDH = lactate dehydrogenase; PFK = phosphofructokinase; PK = pyruvate kinase; HK = hexokinase.

**Table 6 foods-14-03323-t006:** Effects of RP-TAU on energy phosphate metabolites in *M. longissimus thoracis* of heat-stressed crossbred Gan-xi goats.

Items	Groups ^1^		SEM	*p*-Value
	HS	TAU		
PCr (µmol/g)	8.54	8.86	0.537	0.565
Cr (µmol/g)	4.30	6.80	1.320	0.088
ATP (µg/g)	97.7	131.6	10.55	0.009
ADP (µg/g)	351	296	50.90	0.305
AMP (µg/g)	833	284	269.4	0.086
ATP/ADP	0.31	0.48	0.094	0.104
AMP/ATP	9.87	2.24	3.382	0.048

Note: ^1^ HS = 0 kg RP-TAU in basal diet under heat stress; TAU = 0.4% kg/diet RP-TAU in basal diet under heat stress. Energy phosphate metabolites include phosphocreatine, creatine, ATP, ADP, and AMP. RP-TAU = rumen-protected taurine; SEM = standard error of the mean; PCr = phosphocreatine; Cr = creatine; ATP = adenosine triphosphate; ADP = adenosine diphosphate; AMP = adenosine monophosphate.

**Table 7 foods-14-03323-t007:** Effects of RP-TAU on the hepatic gluconeogenesis of heat-stressed crossbred Gan-xi goats.

Items	Groups ^1^		SEM	*p*-Value
	HS	TAU		
Lactate (μmol/gprot)	737	732	88.35	0.955
Glycogen (mg/g)	21.7	22.2	2.687	0.868
Glucose (mmol/g)	11.7	13.5	77.95	0.382
Pyruvate (μmol/gprot)	127	181	31.99	0.148
LDH (U/kgprot)	1.44	1.19	0.202	0.250
PK (U/gprot)	561	589	34.54	0.437
G-6-Pase (ng/mgprot)	3.07	2.89	0.288	0.555
PEPCK (U/kg)	2.30	2.14	0.254	0.564

Note: ^1^ HS = 0 kg RP-TAU in basal diet under heat stress; TAU = 0.4% kg/diet RP-TAU in basal diet under heat stress. RP-TAU = rumen-protected taurine; SEM = standard error of the mean; LDH = lactate dehydrogenase; PK = pyruvate kinase; PEPCK = phosphoenolpyruvate carboxykinase; G-6-Pase = glucose-6-phosphatase.

**Table 8 foods-14-03323-t008:** Effects of rumen-protected taurine on the content of fatty acids in *M. longissimus thoracis* of heat-stressed crossbred Gan-xi goats.

^1^ Items (mg/100 g)	Groups		SEM	*p*-Value
	HS	TAU		
C14:0 Myristic acid	6.80	3.68	1.556	0.137
C16:0 Palmitic acid	171	107	24.51	0.040
C16:1 (n-7) Palmitoleic acid	8.23	4.30	1.414	0.032
C7:0 Heptanoic acid	9.78	6.73	1.856	0.151
C18:0 Stearic acid	208	130	38.54	0.127
C18:1 (n-9) Oleic acid	333	197	54.00	0.083
C18:2 (n-6) Linoleic acid	88.1	69.3	11.51	0.154
C20:3 (n-9) Eicosatrienoic acid	4.50	4.03	0.499	0.378
C20:4 (n-6) Arachidonic acid	47.3	40.6	4.861	0.213
C24:1 (n-9) Nervonic acid	9.45	7.15	0.667	0.014
SFA/UFA	0.81	0.78	0.056	0.635

^1^ HS = (HS + 0 kg RP-TAU in basal diet); TAU = (HS + 0.4% kg/diet RP-TAU in basal diet). SEM = standard error of the mean; SFA/UFA = saturated fatty acids/unsaturated fatty acids.

**Table 9 foods-14-03323-t009:** Effects of rumen-protected taurine on the content of amino acids in *M. longissimus thoracis* of heat-stressed crossed Gan-xi goats.

^1^ Items (g/100 g)	Groups		SEM	*p*-Value
	HS	TAU		
Aspartic acid	1.95	1.94	0.098	0.961
Threonine	0.88	0.86	0.053	0.686
Serine	0.73	0.69	0.040	0.410
Glutamic acid	2.70	2.60	0.160	0.537
Glycine	0.87	0.84	0.056	0.670
Alanine	1.21	1.20	0.067	0.859
Cystine	0.44	0.40	0.037	0.019
Valine	0.90	0.93	0.028	0.555
Methionine	0.23	0.20	0.020	0.184
Isoleucine	0.82	0.85	0.052	0.618
Leucine	1.42	1.46	0.069	0.608
Tyrosine	0.58	0.61	0.036	0.431
Phenylalanine	0.60	0.62	0.040	0.556
Lysine	1.73	1.71	0.087	0.891
Histidine	0.87	0.80	0.061	0.311
Arginine	1.20	1.22	0.070	0.786
Proline	0.71	0.65	0.061	0.327
Umami amino acids	6.24	6.07	0.323	0.618
Essential amino acids	7.42	7.41	0.395	0.995
Total amino acids	17.8	17.6	0.868	0.781

^1^ HS = (HS + 0 kg RP-TAU in basal diet); TAU = (HS + 0.4% kg/diet RP-TAU in basal diet). SEM = standard error of the mean.

**Table 10 foods-14-03323-t010:** Primer sequences of genes.

Gene	Primer Sequence (5′-3′)	Number	Product Size (bp)
*β-actin*	CAGGAAGGAAGGCTGGAAGA	NM_001009784.3	145 bp
	AACATTGGCAGGAAGGGAGA		
*AMPKα2*	CGGGTTGAAGAGATGGAAGC	XR_011257419.1	112 bp
	ACAGTAATCCACGGCAGACA		
*LKB1*	TGGAGTTCAGGATGGAGGTG	XM_027970177.2	132 bp
	TCGATCCGGTGGATGAATGT		
*GSK3β*	ACTACCAAATGGGCGAGACA	XM_069554971.1	167 bp
	GAATCCGAGCATGAGGAGGA		
*GLUT4*	CAGGAAGTGAAACCCAGCAC	XM_027974995.3	145 bp
	CTGTGTGGACCCTCAGTCAT		
*PCK1*	CAGAGAGATACGGTGCCCAT	XM_069548223.1	187 bp
	GGACGTTGAACGCTTTCTCA		
*PCK2*	GCTTGTGGCAAGACAAACCT	XM_069595678.1	178 bp
	ACCTCATCCAGGCAATGTCA		
*PYGM*	GAGCTGGAGGAAATCGAGGA	XM_069566147.1	154 bp
	CATAGCGGATCCCATAGCCA		

Note: *AMPKα2* = AMP-activated protein kinase α2; *LKB1* = receptor serine/threonine kinases 1; *GSK3β* = Glycogen synthesis kinase 3; *GLUT4* = Glucose transporter 4 dehydrogenase; *PCK1* = phosphoenolpyruvate carboxylase 1; *PCK2* = phosphoenolpyruvate carboxylase 2; *PYGM* = glycogen phosphorylase, muscle-associated.

## Data Availability

The original contributions presented in the study are included in the article. Further inquiries can be directed to the corresponding authors.

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
