# Peer review of "Dietary Rumen-Protected Taurine Enhances Growth Performance and Meat Quality in Heat-Stressed Crossbred Gan-Xi Goats via Modulating GLUT4/PYGM-Mediated Muscle Energy Metabolism"

_foods, 2025, doi:10.3390/foods14193323_

Round 1

Reviewer 1 Report

Comments and Suggestions for Authors

Manuscript ID: Foods-3864084. Dietary Rumen-Protected Taurine Enhances Growth Performance and Meat Quality in Heat-Stressed Crossbred Gan-Xi Goats via Modulating GLUT4/PYGM-Mediated Muscle Energy Metabolism

General Comments

The manuscript addresses a relevant and novel topic in ruminant nutrition and heat stress physiology. The findings provide valuable insights with potential applications for livestock production in heat-stressed regions. The manuscript is generally well written; however, several important issues related to methodology, data presentation, statistical analysis, and alignment between results and conclusions must be addressed.

Specific Comments

Title

Lines 1–4. slightly long but generally clear.

Abstract

Lines 13–38. [No further issues identified here.]

Keywords

Lines 39–40. The keywords are sufficient and appropriate.

Introduction

Lines 81–83. Please move the sentence “We hypothesized that… heat stress” to the end of the Introduction to serve as the final paragraph.

Materials and Methods

Lines 89–90. Please clarify whether this study complied with internationally recognized standards for animal welfare and experimental ethics. At least one specific guideline should be cited.

Line 91. Replace “5 mouths of age” with “5 months of age”.

Lines 92–97. For reproducibility, please include the commercial name and manufacturer of the RP-TAU product administered, if applicable. If not, indicate the supplier or source used.

Lines 115–116. Please include the reference to the previous study reporting that a 0.4% dietary TAU supplementation improved heat stress resistance in sheep.

Lines 120–121. Reorganize the table for clarity. The suggested format is: one column listing ingredients, followed below by the list of nutrient contents. On the right, provide a column with the respective values.

Lines 128–131. Present the calculations in a continuous paragraph rather than fragmented text.

Line 137. Please provide the appropriate reference to the “animal welfare guidelines” consulted for this experiment.

Lines 143–146. Remove the redundant sentence “The pH at 45 minutes… textural properties”, as this is already described later in section 2.5. Meat Quality Measurements.

Lines 301–302. The description of the statistical analysis is minimal. Please specify how the data were tested for normality and homogeneity of variances. Were the data normally distributed? Furthermore, the statistical approach is not appropriate: with only two treatments, a t-test is the most direct and suitable method. The General Linear Model (GLM) is not necessary in this context. Additionally, Dunnett’s test is intended for comparing multiple groups to a control, making it inapplicable to the current study.

Results

Lines 304–427. The results are extensive and logically structured. Tables and figures are correctly presented. The results are clearly written and comprehensive.

Discussion

Lines 429–679. The Discussion is generally comprehensive and interprets the key results. However, it often reads as a series of disconnected observations rather than a cohesive narrative. Several sentences are excessively long and complex, making them difficult to follow. Major restructuring and refocusing are needed to strengthen clarity, coherence, and readability.

Conclusions

Lines 681–690. The results mostly support the conclusions. The muscle-related findings are robust and validate the main conclusion.

Lines 692–705. Figure 4 should be removed. It is out of context and is not referenced in either the Results or Discussion sections.

Reviewer 2 Report

Comments and Suggestions for Authors

Article
Dietary Rumen-Protected Taurine Enhances Growth Performance and Meat Quality in Heat-Stressed Crossbred Gan-Xi Goats via Modulating GLUT4/PYGM-Mediated Muscle Energy Metabolism

Keywords:

Keywords: rumen-protected taurine; meat quality; hepatic gluconeogenesis; energy metabolism; heat stress; goat

Replace keywords that are already in the manuscript title 2. Materials and Methods  2.2. Animal Treatments and Experimental Diets Although the treatments are well defined, it is necessary to mention on what basis of nutritional requirements the treatments were defined?. Page 3; Lines 100 to 102“The goats were fed twice daily at 08:00 and 16:00, with water available ad libitum. All uneaten feed was removed and weighed before each morning feeding”.If the feed and leftover amounts were measured daily, you need to specify the criteria for increasing or decreasing the daily amount offered. What level of leftovers was considered?  Figure 1. Temperature-humidity index (THI) during test period.The quality of figure 1 needs to be improved, especially the formatting (figure size, font size, sharpness, etc.) 3. Results3.2. Growth Performance and Viscera IndexTable 2. Effects of rumen-protected taurine on the growth performance and viscera index of heat stressed crossbred Gan-xi goats.

Items

HS

TAU

SEM

P-value

ADG (g/d)

0.15

0.19

0.007

0.012

Average daily gain is expressed in grams per day or kilograms per day  3.4. M. Longissimus Thoracis Composition and Meat QualityTable 4. Effects of RP-TAU on the M. longissimus thoracis composition and meat quality of heat stressed crossbred Gan-xi goats.

Items

HS

TAU

SEM

P-value

Hardness (kg)

Gumminess (kg)

Check the unit of table items?

Comments on the Quality of English Language

I don't have the necessary knowledge to evaluate the English. I suggest having it reviewed by a native speaker

Round 2

Reviewer 1 Report

Comments and Suggestions for Authors

The authors have successfully addressed the requested changes.
This reviewer has two additional general comments:

1. Consider reducing and standardizing the number of decimal places in the tables for easier reading. We recommend using numeric formatting such as 000, 00.0, 0.00, or 0.000.

2. The paragraphs in the discussion are too long, making them difficult to read. We suggest breaking up these lengthy paragraphs so that each one focuses on a single main idea.
